# OPTO-BLUE: An Integrated Bidirectional Optogenetic Lentiviral Platform for Controlled Light-Induced Gene Expression

**DOI:** 10.3390/ijms24119537

**Published:** 2023-05-31

**Authors:** Duxan Arancibia, Iracy Pol, Martín Vargas-Fernández, Rafaella V. Zárate, Janetti R. Signorelli, Pedro Zamorano

**Affiliations:** 1Departamento Biomédico, Facultad de Ciencias de la Salud, Universidad de Antofagasta, Antofagasta 1240000, Chile; duxan.arancibia@uantof.cl (D.A.); iracyesther@gmail.com (I.P.); martin.vargas.fernandez@gmail.com (M.V.-F.); rafaella.zarate@gmail.com (R.V.Z.); janetti.signorelli@uantof.cl (J.R.S.); 2Instituto Antofagasta, Universidad de Antofagasta, Antofagasta 1240000, Chile

**Keywords:** optogenetic, lentivirus, GAVPO, gene-regulated expression, bidirectional

## Abstract

Regulated systems for transgene expression are useful tools in basic research and a promising platform in biomedicine due to their regulated transgene expression by an inducer. The emergence of optogenetics expression systems enabled the construction of light-switchable systems, enhancing the spatial and temporal resolution of a transgene. The LightOn system is an optogenetic tool that regulates the expression of a gene of interest using blue light as an inducer. This system is based on a photosensitive protein (GAVPO), which dimerizes and binds to the UASG sequence in response to blue light, triggering the expression of a downstream transgene. Previously, we adapted the LightOn system to a dual lentiviral vector system for neurons. Here, we continue the optimization and assemble all components of the LightOn system into a single lentiviral plasmid, the OPTO-BLUE system. For functional validation, we used enhanced green fluorescent protein (EGFP) as an expression reporter (OPTO-BLUE-EGFP) and evaluated the efficiency of EGFP expression by transfection and transduction in HEK293-T cells exposed to continuous blue-light illumination. Altogether, these results prove that the optimized OPTO-BLUE system allows the light-controlled expression of a reporter protein according to a specific time and light intensity. Likewise, this system should provide an important molecular tool to modulate gene expression of any protein by blue light.

## 1. Introduction

Regulated gene switches were designed and constructed to turn “on” and “off” the expression of a transgene using an inducer [1,2]. The development of these switches seeks to properly dose the delivery of a therapeutic gene product (RNA or protein), avoiding the toxic effects of overexpression when constitutive promoters are used. This feature makes the regulatable gene switches useful in basic research for studies requiring especially precise and acute control of the gene delivery to unveil its function. Furthermore, gene switches have promising uses in biomedicine to provide regulated dosed delivery of biopharmaceuticals or for use in gene replacement therapy, as reviewed in [3].

A critical issue for gene switches is the choice of the appropriate inducer to provide high spatial and temporal resolution of the expression. Most of the regulatable gene switches initially used chemical inducers [4], but they can be toxic in high doses and may freely diffuse in the system, being difficult to control its location or elimination and preventing the fine-tuning required for therapeutic gene expression. These features limit the temporal and spatial resolution of chemical-induced gene expression [5]. Light-activated gene switches have been proposed as an alternative to chemical gene switches [6,7,8,9,10]. Light has proven to be an ideal inducer to overcome chemical inducers, as it is cheap, easy to obtain, highly adjustable, and renders higher spatial and temporal control of expression [11,12,13].

Distinct optogenetic gene switches have been developed to regulate transgene expression using different wavelengths and light-responsive proteins [14,15]. The LightOn system is an optogenetic gene switch developed by Wang. et al. [16], composed of two components: (i) a blue light switchable chimeric transcription factor GAVPO containing a photosensitive LOV domain, which homodimerizes upon exposure to blue light, and (ii) a GAVPO responsive promoter composed of five repetitions of the upstream activation sequence of Gal4 (UAS), fused to a minimal adenovirus E1b promoter (UAS inducible promoter). The LightOn system has been shown to be able to control transgene expression in HEK293 cells and mice [14]. In addition, it has been used to control the cell fate of neural progenitor cells and chick-limb mesenchyme cells [17,18], study cell ablation in zebrafish [19], regulate blood-glucose homeostasis in type 1 diabetic mice [20] and target a potential breast cancer tumor by a light-switchable gene expression system encapsulated in a nanoparticle delivery system [20].

We previously optimized and adapted the LightOn system to a lentiviral platform [21]. Lentiviruses allow the integration of genetic material into the host-cell genome so that genetic incorporation remains stable over time [22]. The lentiviral plasmids are also an advantageous tool for manipulating the genomes of post-mitotic cells, such as neurons. In our previous work, we constructed two lentiviral plasmids bearing the two components of the LightOn system: (i) one that expresses GAVPO under the Ubiquitin constitutive promoter; and (ii) a second plasmid that expresses a gene of interest (GOI) driven by the UAS inducible promoter. We demonstrated the functionality of this system through the blue light-induced expression of the cerebral dopamine neurotrophic factor and mCherry by transfection and transduction of mammalian cell lines and neurons [21]. As the LightOn lentiviral system is based on two lentiviruses, its efficient operation requires the co-transduction of the target cells in the tissue where the regulation of gene expression is desired. This co-transduction requirement decreases the efficient and equitable delivery of all LightOn system components due to possible interference between the lentiviruses to achieve the dual transduction of the target cells. To overcome this problem, we designed and constructed the OPTO-BLUE system, an integrated bidirectional lentiviral system containing all the components of the LightOn system in a single lentivirus, thus improving the mentioned disadvantages and favoring the equitable incorporation of all the components through a single transduction event. We constructed the pOPTO-BLUE-EGFP bidirectional lentiviral plasmid and optimized its design to minimize background activity in darkness. We transfected pOPTO-BLUE-EGFP in HEK293-T mammalian cells to evaluate the functionality of the OPTO-BLUE system and the optimal lighting parameters to achieve an efficient expression. The results showed that blue light-induced enhanced green fluorescent protein (EGFP) level expression could be regulated through intensity and lighting time exposure, reaching a two-fold induction after 16 h of illumination. We further demonstrated blue light-inducible EGFP expression by transducing HEK293-T cells. These findings prove that the OPTO-BLUE system allows the blue light control of a GOI in mammalian cells, suggesting its potential use in biomedical applications requiring dosed therapeutic gene expression.

## 2. Results

### 2.1. Design and Construction of the OPTO-BLUE System

Previously, we optimized and adapted the LightOn system to a lentiviral platform [21]. Although this system allows for the blue light-inducible expression of a GOI in mammalian cells, it is a dual system based on two lentiviral transfer plasmids, the pFUG-1D/2A-HA-GAVPO-W and pF-UAS(s)-mCherry-W plasmids, containing the two main components of the LightOn system: the light-switchable transcription factor GAVPO and the UAS inducible promoter to which it binds, which contains five consecutive repeats of the upstream activating sequence of Gal4 (5xUASG) fused to a TATA sequence of the minimal promoter E1b (5xUASG-TATA). The functionality of this two-vector design requires the transduction of the target cells with both vectors to obtain an operational light-inducible expression system. This requirement reduces the efficiency and applications of the LightOn system, where light-regulated gene expression is desired.

We designed and constructed an integrated OPTO-BLUE system that carries all components of the LightOn system assembled in a single, bidirectional, lentiviral transfer plasmid to regulate the expression of the GOI. The bidirectional design of the OPTO-BLUE system allows, in one direction, the expression of the cis-activating GAVPO protein placed downstream of an internal constitutive Elongation Factor 1α short (EFS) promoter, and upstream in the opposite direction, the UAS inducible promoter driving the expression of the GOI. Thus, when the GAVPO protein is exposed to blue light, it dimerizes and binds to the UAS inducible promoter, activating GOI expression (Figure 1A).

### 2.2. Optimization and Proof of Concept of the OPTO-BLUE System

We constructed the pOPTO-BLUE-EGFP-V1 plasmid by inserting the EGFP sequence into the pOPTO-BLUE plasmid to evaluate the functionality of the OPTO-BLUE system (Figure 1B). We generated the pOPTO-BLUE-EGFP-V2 plasmid by replacing the Ubiquitin promoter with the EFS promoter (Figure 1B). The EFS promoter is a much shorter promoter compared to the Ubiquitin promoter, allowing a decrease in the size of the packaged viral transcript from 7.2 to 6.2 kb. To determine differences in the inducible expression of EGFP by both vectors driving the expression of GAVPO with the Ubiquitin and EFS promoters, HEK293-T cells were transfected with pOPTO-BLUE-EGFP-V1 or -V2 (Figure 2A). To activate GAVPO and induce EGFP expression, cell cultures were exposed to blue light for 16 h. The expression of EGFP was detected by immunoblot using an anti-GFP antibody (Figure 2B). The quantification of EGFP protein levels showed a two-fold increase in cell cultures transfected with pOPTO-BLUE-EGFP-V1 and exposed to blue light, compared to dark conditions (Figure 2C, F(1, 8) = 36.28; *p* = 0.0029). Similar increases in EGFP protein levels were achieved by employing the EFS promoter (Figure 2C, F(1, 8) = 36.28; *p* = 0.0107). These results indicate that the EFS promoter, which is six times smaller than the Ubiquitin promoter, drives the expression of a GOI with similar efficiency.

### 2.3. Optimization of the Background Expression of the OPTO-BLUE System

As is shown in Figure 2B, with both pOPTO-BLUE-EGFP-V1 and -V2 plasmid, a significant background expression of EGFP is also detected in cell cultures maintained in the darkness conditions. We hypothesized this background expression could be due to the close proximity between both the constitutive (Ubiquitin or EFS) and the UAS inducible promoters. To test whether the close proximity of the promoters is increasing the background expression in darkness, we inserted different sizes of a stuffer-spacer sequence between the EFS and the UAS inducible promoter, expanding the physical distance from each other (Figure 3A). Four spacer sequences with different sizes (200, 500, 1000, and 2000 bp) were inserted into the pOPTO-BLUE-EGFP -V2 plasmid, generating the pOPTO-BLUE-EGFP-V3_200bp, pOPTO-BLUE-EGFP-V3_500bp, pOPTO-BLUE-EGFP-V3_1000 bp and pOPTO-BLUE-EGFP-V3_2000bp plasmids (Figure 3B). The different plasmids with spacers were transfected in HEK293-T cells and maintained in darkness for 24 h, and then, the EGFP expression was assessed by an immunoblot assay (Figure 3C). The quantification of EGFP protein levels showed a slight EGFP background expression in cells transfected with pOPTO-BLUE-EGFP-V2 (Figure 3D, control). No significant differences were detected in EGFP protein levels between cells transfected with pOPTO-BLUE-EGFP-V2 and pOPTO-BLUE-EGFP-V3_200bp plasmids (Figure 3D, F(4, 10) = 30.91; *p* = 0.9838). Nevertheless, a significant decrease in EGFP background expression was observed in cells transfected with pOPTO-BLUE-EGFP-V3_500bp, 1000bp, or 2000bp plasmids compared to the pOPTO-BLUE-EGFP-V2 plasmid (Figure 3D, F(4, 10) = 30.91; 500 bp: *p* = 0.0029, 1000 bp: *p* = 0.0012 and 2000 bp: *p* = 0.0004).

We also hypothesized whether the background expression in the OPTO-BLUE system could be partly due to a background activation of GAVPO, as has been previously reported [22]. To prove this idea, we constructed a version of pOPTO-BLUE-EGFP-V2 without the EFS constitutive promoter that drives the expression of GAVPO protein (Figure 4A). HEK293-T cells were transfected with OPTO-BLUE-EGFP-V1, -V2, -V2 without EFS promoter (W/O EFS), and -V3_500bp plasmids were constructed to compare the inducible and background expression of EGFP in cell cultures exposed to blue light and darkness conditions, respectively. The quantification of the EGFP expression detected by immunoblot showed the expected background expression using pOPTO-BLUE-EGFP-V1 and -V2 plasmids (Figure 4B,C, F(3, 16) = 3.279; *p* > 0.9999). Interestingly, this background expression is reduced in cells transfected with a pOPTO-BLUE-EGFP-V3_500bp plasmid (Figure 4B,C, F(3, 16) = 3.279; *p* < 0.0001), which is directly related to the concentration of transfected DNA (Appendix A). Remarkably, the background expression of EGFP was completely removed without the promoter of the GAVPO cassette, suggesting that the constitutive EFS promoter influences the background expression of EGFP in the absence of illumination (Figure 4B,C, F(3, 16) = 3.279; *p* < 0.0001).

Together, these results demonstrate the importance of the physical separation of promoters in a bidirectional design of gene switches. For the OPTO-BLUE system, 500 bp or more between the constitutive and inducible promoters are sufficient to reduce background expression. Moreover, the background expression is partly due to a combination of the proximity of the promoter and the activation of GAVPO in darkness.

### 2.4. Evaluation of the Optimal Blue Light Intensity and Exposure Time of the OPTO-BLUE System

We assessed different times of exposure and intensities of blue illumination to determine the appropriate parameters to obtain optimal induction efficiency with OPTO-BLUE and to demonstrate a tunable capacity of the GOI expression. First, we transfected HEK293-T cells with pOPTO-BLUE-EGFP-V3_500bp plasmid and induced with blue light the expression of EGFP at different time windows (0, 1, 2, 4, 6, 8, 9, and 24 h). The time-course expression of EGFP was assessed by immunoblot (Figure 5A). We detected faint bands of EGFP expression after 2 h of illumination and more intense bands after 6 h of induction. The quantification of EGFP expression reveals a significant increase in EGFP protein expression from 8 h of lighting onwards compared to the initial condition, 0 h (Figure 5B, F(7, 16) = 20.87; 8 h: *p* = 0.0077, 9 h: *p* = 0.0001 and 24 h: *p* ≤ 0.0001). We also evaluated whether the expression of EGFP is tunable with lower blue-light intensities at 20, 40, and 80 lux rather than the standard 160 lux used in all experiments (Figure 5C). The quantification of EGFP expression shows a significant decrease of EGFP expression at 20 and 40 lux, suggesting that this system could be regulated by light intensity (Figure 5D, F(3, 8) = 4.867; 160 vs. 20 lux: *p* = 0.0192, 160 vs. 40 lux: *p* = 0.0457). Together, these results provide evidence that the inducible expression of the GOI in the OPTO-BLUE system can be regulated by adjusting the light illumination intensity and time exposure.

### 2.5. Evaluation of the OPTO-BLUE System by Lentiviral Particles in HEK293-T Cells

A remarkable feature of lentiviruses is their ability to integrate into the host genome providing long-term expression. To verify the functionality of the OPTO-BLUE system through transduction, we produced lentiviral particles of the pOPTO-BLUE-EGFP-V3_500bp plasmid and transduced HEK293-T cells (see Materials and methods for details). We also transfected HEK293-T cells with pOPTO-BLUE-EGFP-V3_500bp plasmid to compare the EGFP expression levels obtained with transduction and transfection methods. After a blue-light induction protocol, we evaluated the background and inducible expression of EGFP through immunoblotting. The immunoblot shows that both conditions, transduced and transfected exhibited a significant increase in EGFP expression after illumination (Figure 6A,B, F(1, 8) = 61.63; transfection: *p* = 0.0013 and transduction: *p* = 0.0009). Interestingly, the transduced cell cultures exhibited a lower background expression in the dark condition compared with the transfected cells.We also verified the expression of GAVPO by immunoblot using an anti-HA antibody, which is expressed in both transfected and transduced cells (Figure 6A). Finally, we observed light-induced EGFP expression in HEK293-T cells transduced with the lentivirus generated from pOPTO-BLUE-EGFP-V3_500bp. Cells were exposed to darkness and blue light. We observed cells transduced and expressing EGFP in blue-light conditions. However, in dark conditions, we were not able to detect EGFP expression (Figure 6C). This result is in agreement with the immunoblot, where a thin band of EGFP expression is observed in dark conditions (Figure 6A).

## 3. Discussion

The Light-On inducible expression system is a highly regulated, light-inducible expression system with multiple applications in biology. Accordingly, its therapeutic approach could be of ample use in gene therapy [11,14]. This system allows spatiotemporal control of gene expression, where light could tightly control the dosed expression of therapeutic proteins or biologicals. For the effective use of this system in gene therapy, especially in the central nervous system, it is required to adapt the molecular components into the proper vector that efficiently could incorporate the Light-On system into the target cells or tissues. For that purpose, we have adapted the Light-On system to a single lentiviral platform that has shown to be very reliable and efficient in transferring genetic information in HEK293-T cells. We previously developed a lentiviral Light-On platform that relies on the co-transduction of two viral particles to regulate gene expression effectively [21]. Although this system is appropriate in an experimental setting in cell biology research, it is not practical for applications in a therapeutic situation, due to the difficulty of transducing two viral particles to the same target cell, where the insertion of the expression cassettes of the GAVPO transactivator and the regulated light-induced UAS expression cassette with the GOI is required. This is much easier achieved with a two-plasmid system carrying each expression cassette, as recently demonstrated by the inclusion of both plasmids in a nanoparticle delivery system for the expression of diphtheria toxin for cancer treatment, where a spatiotemporal control is achieved [20]. Here, we showed a functional and inducible Light-On system in a single lentiviral vector, coined OPTO-BLUE, where the cassette for the constitutive expression of the transactivator protein GAVPO and the GAVPO-regulated UAS minimal-promoter driving the expression of the GOI is arranged in a head-to-head configuration. This arrangement allows the generation of lentiviral particles fully capable of transducing mammalian cells carrying the LightOn system. Furthermore, we characterized the optimal separation between the promoters to decrease background expression in the absence of light, as well as the light-intensity and temporal induction of a reporter protein EGFP on this system.

The OPTO-BLUE lentiviral system is a single integrated Light-On inducible expression system that combines the cassette for the expression of the GAVPO transactivator and the UAS cassette containing the GOI in a single viral transcript. Different hurdles must be overcome and tested to demonstrate that a functional and inducible lentiviral vector was obtained. We first attempted to compact most of the genetic elements required for GAVPO expression, reducing the size of the constitutive Ubiquitin promoter for the EFS promoter. The EFS promoter is robust and resistant to the silencing of the promoter in a lentiviral setting [23], allowing a reduction of approximately 1 kb of the packaged viral transcript. Although the size of the maximal packaged lentiviral genome has not been defined, it is known that increasing the size of the transcript results in a lower viral titer [24]. Therefore, the reduction of approximately 1 kb on the packaging transcripts should not only allow the expression of a larger GOI, but also could result in higher titers for those small viral particles that harbor smaller genomes. Furthermore, the decrease in the transfer plasmid size should also lead to better transfection efficiency during viral production, helping to obtain better viral titers [25].

The mechanism of gene activation in the OPTO-BLUE system involves the dimerization of GAVPO, that is, transduced cells should be acting on a single site or a few sites, if considered a single or a few viral-integration events. Although the dynamics of GAVPO gene activation are not completely resolved, the LOV domain has an equilibrium–dissociation constant (Kd) in a micromolar range [26], suggesting that a robust promoter must drive the expression of GAVPO. The EFS promoter reached similar levels of GAVPO expression to the Ubiquitin promoter. Therefore, the OPTO-BLUE system with the EFS promoter should achieve comparable functionality as the Ubiquitin promoter.

The second improvement to address was to design a transfer vector with both elements of the Light-On system as a pair of cis-acting genetic elements located on different expression cassettes. This design should permit the synthesis of full transfer/packaging transcripts containing both long terminal repeats (LTRs) in a configuration that allows the entire retro transcription process. This allows the insertion and transduction of eukaryotic cells by the lentiviral particles. The generation of the packaging transcript in many lentiviral systems is driven by a CMV promoter located upstream of the 5′LTR sequence. RNA pol II is recruited by this promoter, generating a capped transcript that is polyadenylated at the 3′ ends by the polyadenylation signal on the 3′LTR [27]. The two components of the Light-On system consist of two expression cassettes, one for the expression of GAVPO and the second for the expression of GOI. Both cassettes are driven by RNA pol II; therefore, both cassettes must contain a polyadenylation signal. We hypothesize that the best arrangement of both expression cassettes, acting in *CIS*, without interfering with the generation of a full transcript, was by designing the UAS-GOI-polyA signal expression cassette in the opposite direction of the EFS-GAVPO-LTR (polyA signal) cassette. The polyA signal in the UAS-GOI-polyA cassette is in the opposite strand and should not interfere with generating the full package transcript. Transfection experiments indicate that both cassettes are functional, as EGFP expression is activated by light, and the GAVPO protein is detected. Despite this functionality, we observed a high-background expression in the darkness. Considering the minimal UAS promoter is of short size, and the EFS promoter is a robust promoter [23], the head-to-head configuration of both promoters could result in a high-background expression due to the closeness of both promoters, in a manner similar to the design of bidirectional Light-On vectors [12]. Therefore, we constructed a series of OPTO BLUE vectors with different spacers to assess the background expression of the GOI. The stuffer fragment was amplified from ORF of Cas9 using the pLenti-CRISPR-Cas9-P2-mCherry-V2 vector and comprehensively analyzed to avoid the presence of regulatory gene elements. As expected, the background expression was decreased as a measure of the distance separating the promoters, highly suggesting that the assembly of the transcription initiation complex of the EFS promoter could be inducing the expression of the GOI. We tried stuffer sequences of different sizes, from 0.2–2.0 kb, and found that a separation of 0.5 kb is enough to decrease background expression significantly. The background expression is even lower with larger spacers, obtaining the lowest background with the largest spacer tested at 2.0 kb. However, we suggest that a spacer of 0.5 kb should be enough to be functional, offering a significant signal (light) to noise (background) ratio; but the interplay between the amount of background desired and the size of the GOI wanting to be expressed should be considered when choosing the proper OPTO BLUE plasmid.

The Light-On system is quite sensible for light induction [11]. The OPTO BLUE system seems to maintain this dynamic since it can increase the expression of the GOI after 2 h of induction, which becomes significant after 8 h of illumination. This expression keeps increasing as the light induction is maintained, similar to previous reports [14,21]. Furthermore, we observed that the expression of EGFP could be regulated by light intensity. Similar to the LuminOn system, the OPTO-BLUE system could allow pulsatile and quantitative activation of transgene expression in a light irradiance-dependent manner [28]. Even a low intensity seems to be enough to activate the OPTO-BLUE system. The low blue-light intensity can induce the covalent cysteinyl-flavin adduct, which allows the dimerization of GAVPO [29]. This characteristic has not been fully studied, but it is important for the therapeutic applications of this technology, since the lower the intensity required for gene activation, the less energy will be required by the light-generating devices to be utilized in a clinical setting.

To our knowledge, this is the first, fully integrated, lentiviral Light-On vector system to be functional. The arrangement of the genetic elements permits the induction of the GOI with a low-background expression. The improved Light-On platform described here is an improved version of the two-component lentiviral platform developed previously [21], that simplified its use in many biological settings. Despite this simplification, some background expression in the absence of light is observed that could be avoided using a larger separation between the promoters. We suggest that a 0.5 kb separation should be optimal for most of the GOI to be used, but when a tighter control of the background is required, a larger separation of 2.0 kb is recommended. This configuration could be used with the improved versions of GAVPO [30], permitting a lower background expression that could be used in an experimental or therapeutic setting.

## 4. Materials and Methods

### 4.1. Lentiviral Plasmid Construction

We previously developed pFUG-1D/2A-HA-GAVPO-W (12 kb) plasmid that expresses EGFP and GAVPO under the Ubiquitin constitutive promoter, and pF-UAS(s)-mCherry-W (10 kb) plasmid that expresses mCherry under the 5xUASG-TATA (UAS) inducible promoter. Standard molecular cloning and recombinant DNA techniques were used for plasmid construction. Lentiviral OPTO-BLUE expression plasmids were assembled from the FUGW plasmid [22], as detailed below: the GAVPO sequence was obtained from pGAVPO plasmid using *PacI* and *BsrGI* restriction enzymes (New England Biolabs, Ipswich, MA, USA). Then, the EGFP sequence from the FUGW plasmid was replaced by the GAVPO sequence to obtain pFU-GAVPO-W (11 kb). This plasmid was digested with *NspI* enzyme (New England Biolabs, Ipswich, MA, USA) to reduce the size. The reduced version of pFU-GAVPO-W (psFU-GAVPO-W) was achieved by releasing expendable sequences (~2 kb). The 9 kb psFU-GAVPO-W plasmid was used to construct the first version of pOPTO-BLUE. The UAS inducible promoter with a multiple cloning site (UAS-MCS) was obtained from pUAS-mcherry plasmid by PCR using the following primers sets flanked by *AsiSI* and *PacI* restriction sites: FP_AsiSI_UAS: 5’-GCGTTAATTAAAAGTGCAGGTGCCAGAAC-3′ and RP_PacI_UAS: 5′-CGCGCCGCGGGCGATCGCTAAGATACATTGATGAGT-3′. Then, the psFU-GAVPO-W plasmid was linearized using the *PacI* enzyme (New England Biolabs, Ipswich, MA, USA), and the UAS-MCS amplicon previously treated with the *AsiSI* and *PacI* enzymes were introduced. A restriction-enzyme assay was performed to corroborate the insert orientation, obtaining the pOPTO-BLUE bidirectional plasmid. This plasmid expresses GAVPO driven by the Ubiquitin promoter in one direction and a GOI driven by the UAS inducible promoter in the opposite direction. From the pOPTO-BLUE plasmid, the pOPTO-BLUE-EGFP plasmids were constructed. To achieve this, the coding sequence for EGFP was obtained from pFUGW plasmid by PCR using the following primer sets flanked by *AscI* and *BstBI* restriction sites: FP_AscI_GFP: 5′-ATATGGCGCGCCATGGTGAGCAAGG-3′ and RP_BstBI_GFP: 5′-CGCGTTCGAACTACTTGTACAGCTCGT-3′. Then, the EGFP amplicon was introduced between *AscI* and *BstBI* restriction sites in the pOPTO-BLUE plasmid to obtain pOPTO-BLUE-EGFP-V1.

To reduce the size of the pOPTO-BLUE-EGFP-V1 plasmid even more, the Ubiquitin promoter was replaced by the EFS promoter to obtain pOPTO-BLUE-EGFP-V2. The EFS sequence was obtained by PCR from the pLenti-CRISPR-Cas9-P2A-mCherry plasmid using the following sets of primers: FP_EFS: 5′-ATATTTAATTAATGGCTCCGGT-3′ and RP_EFS: 5′-TGGCAGCGCTCTAGAACCGGT-3′. The EFS amplicon was treated with *AgeI* and *PacI* enzymes (New England Biolabs, Ipswich, MA, USA) and introduced into the pOPTO-BLUE-V1 plasmid between *AgeI* and *PacI* restriction sites. The new plasmid version, constructed with an EFS promoter, was named pOPTO-BLUE-EGFP-V2. Moreover, a pOPTO-BLUE-EGFP-V2_W/O plasmid without Ubiquitin/EFS promoter was generated. Since the Ubiquitin sequence is flanked by *PacI* sites, its extraction was performed by enzymatic digestion with *PacI*. Then, the ligation of their compatible ends was performed. Finally, different sizes of spacer sequences were introduced into the pOPTO-BLUE-EGFP-V2 plasmid to generate pOPTO-BLUE-EGFP-V3. DNA spacers of 200, 500, 1000, and 2000 bp were obtained from pLenti CRISPR-Cas9-P2A-mCherry, using the following set of primers: FP_200bp: 5′-CGCTTAATTAAGCTGTACGAGTACTTCAC-3′ and RP_200bp: 5′-CGCTTAATTAAAGATTTCCACGGAGTCGA-3′; FP_500bp: 5′-CGCTTAATTAAGCTGTACGAGTACTTCAC-3′and RP_500bp: 5′-CGCTTAATTAAAGATTTCCACGGAGTCGA-3′; FP_1000bp: 5′-CGCTTAATTAAGCTGTACGAGTACTTCAC-3′ and RP_1000bp: 5′-CGCTTAATTAAGTCGATGGAGTCGTCCT-3′; FP_2000bp: 5′-CGCTTAATTAAGCTGTACGAGTACTTCAC-3′ and RP_2000bp: 5′-CGCTTAATTAACTCGAAGCTGCTTCTTTC-3′). The plasmid versions obtained were pOPTO-BLUE-EGFP-V3_200bp, pOPTO-BLUE-EGFP-V3_500bp, pOPTO-BLUE-EGFP-V3_1000bp, pOPTO-BLUE-EGFP-V3_2000pb (pOPTO-BLUE-EGFP-V3 + spacers).

### 4.2. Cell Culture and Transfection

HEK293-T cells were cultured in DMEM (Dulbecco’s modified Eagle’s medium), supplemented with 10% (*v*/*v*) fetal-bovine serum, 100 U/mL penicillin, and 100 μg/mL streptomycin and L-glutamine (2 mM). All culture reagents were from Gibco. Cell cultures were maintained at 37 °C in an atmosphere of 95% air and 5% CO_2_. Cells were transfected with a Calfectin agent following the manufacturer’s recommendations (Calbiotech, El Cajon, CA, USA) using the following plasmid: psFEGW, pOPTO-BLUE-EGFP-V1, pOPTO-BLUE-EGFP-V2, pOPTO-BLUE-EGFP-V2_W/O, and pOPTO-BLUE-EGFP-V3 + spacers.

### 4.3. Lentivirus Production and Transduction

HEK293-T cells were transfected with Lipofectamine 2000 (Thermo Fisher Scientific, Waltham, MA, USA) using the packaging plasmids pCMV-ΔR8.9, pCMV-VSVg, and the corresponding transfer plasmid pOPTO-BLUE-EGFP-V3_500bp to produce a functional lentivirus, as described previously [22]. HEK293-T cells were grown in Dulbecco’s Modified Eagle Medium DMEM supplemented with 10% fetal-bovine serum and 100 U/mL penicillin, and 100 μg/mL streptomycin (Thermo Fisher Scientific, Waltham, MA, USA). Lentiviral particles were collected from the supernatants 48 h after transfection and the lentiviral particles were centrifuged, passed through a 0.45-μm filter, and stored at −80 °C. Briefly, transduction was performed by adding 250 μL of lentiviral particles to HEK293-T cells, previously cultivated in 24 well plates. HEK293-T cell transduction was determined by fluorescence analysis and Western blot of EGFP.

### 4.4. Blue Light Induction Assay

After transfection or transduction, a group of plates with HEK293-T cells was maintained in darkness, while other groups of plates were exposed to blue light using a light intensity of 160 lux at 37 °C with 5% CO_2_. For all the induction experiments (except where indicated), cells were transfected for 8 h followed by 16 h of continuous illumination with blue light (LED, 460 nm). To perform time exposure of blue light experiments, 8 h post-transfection, different plates were exposed to continuous blue-light illumination for 0.25, 0.5, 1, 2, 4, 6, 8, 9, or 24 h. For light-intensity experiments, cells were transfected for 8 h and induced with blue light for 16 h at different light intensities (20, 40, 60, 80, and 160 lux). Finally, for transduction experiments, 76 h after transduction with lentiviruses, cells were exposed to 16 h of continuous illumination with blue light.

### 4.5. Western Blotting Analysis

For immunoblotting experiments, cell cultures were homogenized with a lysis buffer (50 mM sodium acetate, 150 mM sodium chloride, 10% glycerol (*v*/*v*), dH_2_O) containing protein inhibitors, then the lysate was centrifuged at 10,000× *g* for 5 min to remove debris. The lysate supernatant was preserved at −80 °C. Protein concentration was estimated by the DC protein assay method (Bio-Rad). The equivalent of 30 µg of protein extract was separated by SDS-PAGE and then transferred to a nitrocellulose or polyvinylidene difluoride (PVDF) membrane (GE Healthcare Bio-Sciences Corpstate, Piscataway, NJ, USA). Membranes were treated with a blocking buffer (3% Bovine Serum Albumin (BSA)) for 1 h at room temperature (RT). Then, membranes were probed with the appropriate primary antibodies, 2 h at RT for anti-GAPDH or overnight at 4 °C for anti-GFP. Then, the membranes were washed with a Tris Buffer Solution with 0.1% Tween (TBS-T buffer) and incubated with HRP-conjugated secondary anti-mouse/rabbit IgG antibodies (1:5000, Li-COR) for 1 h at RT. The blots were revealed by an enhanced ECL chemiluminescence system (LICOR detection system) and detected by a C-Digit Blot Scanner (Li-COR). Bands were quantified by densitometry using Fiji software Version 1.53t24 (NIH, Bethesda, MD, USA). The primary antibodies included an anti-GFP rabbit polyclonal antibody (A-11122, 1:2000, Thermo Fisher Scientific, Waltham, MA, USA), anti-GAPDH mouse monoclonal antibody (sc-32233, 1:2000, Santa Cruz Biotechnology, Dallas, TX, USA), and anti-HA mouse monoclonal antibody (sc-7392, 1:2000, Santa Cruz Biotechnology, Dallas, TX, USA) [31].

### 4.6. Epifluorescence Microscopy Imaging

Cells were fixed using 4% paraformaldehyde for 10 min at room temperature. After fixation, three washes with PBS 1× solution were performed, and a final wash in distilled water was made to remove excess salt. Finally, the coverslips were mounted with a Vectashield/DAPI solution (Vector). Images were acquired in dark and light conditions with a DS-Fi2 epifluorescence microscope (Olympus, Shinjuku-ku, Tokyo, Japan) equipped with a Nikon DS-Fi2 camera (Nikon, Minato-ku, Tokyo, Japan) operated with QCapture Suite PLUS 3.1.3.10 (Q-Imaging).

### 4.7. Statistical Analysis

The data obtained with the Fiji program were analyzed and processed with the GraphPad Prism 8 program. The values shown in the figures represent the mean ± SEM (Standard Error of the Mean) results obtained from at least three independent experiments. The Shapiro–Wilk test for normality was used to determine whether parametric or nonparametric statistical tests were to be used. When two or more experimental groups with one factor were compared, the one-way analysis of variance (“one-way ANOVA”) was used, followed by Dunnet’s multiple-range comparison as a post-hoc test. A two-way ANOVA with Dunn’s analysis as a post-hoc test was used to compare data of multiple groups with two independent factors. The differences between the experimental groups were considered statistically significant from a confidence level greater than 95% (*p* < 0.05).

## 5. Patents

A PCT filing for the OPTO-BLUE system has been requested. PE889PCTORCL.

## Figures and Tables

**Figure 1 ijms-24-09537-f001:**
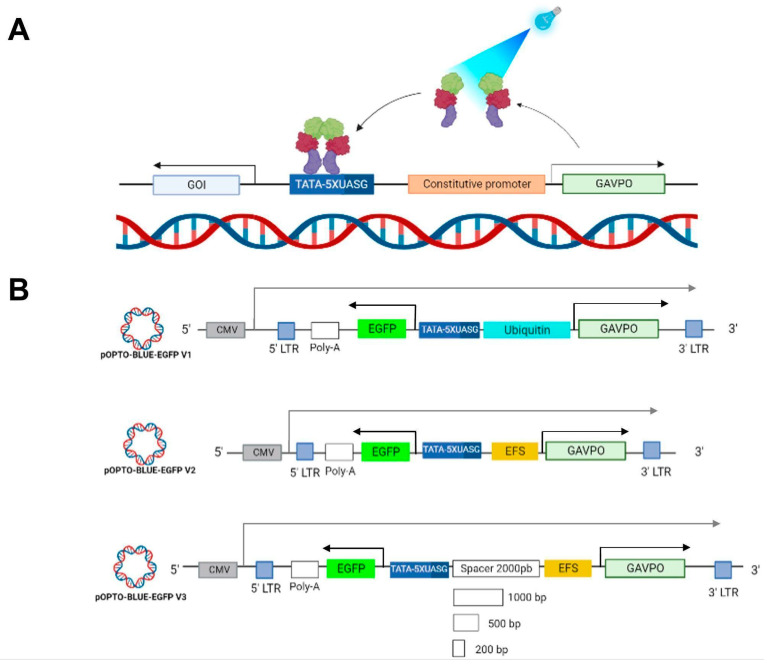
OPTO-BLUE gene expression system and designed vectors. (**A**) Arrangement of LightOn gene expression system developed by Wang & Cols. to regulate gene expression by blue light in the OPTO-BLUE system. This system comprises (1) a GAVPO protein regulator (light-green box), which activates under blue light (460 nm) exposure, binding to s (2) a regulator sequence, the UAS inducible promoter (blue box), promoting s (3) the gene/protein expression of interest (GENE OF INTEREST, GOI). (**B**) Schematic illustration of the LightOn lentiviral plasmids constructed: pOPTO-BLUE-EGFP-V1, pOPTO-BLUE-EGFP-V2, and pOPTO-BLUE-GFP-V3 + spacers. Black arrows indicate the transcription start site for EGFP (as the GOI) and GAVPO expression driven by the inducible and constitutive promoter, respectively. The gray arrow indicates the transcription start site for the viral transcript driven by the CMV promoter.

**Figure 2 ijms-24-09537-f002:**
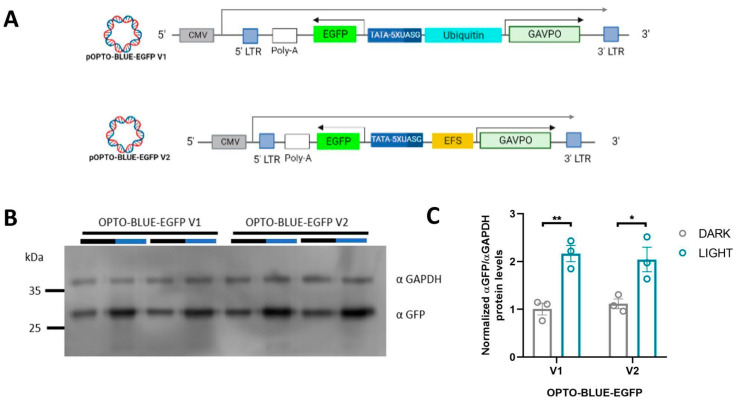
pOPTO-BLUE lentiviral vector can induce EGFP expression in HEK293-T cells by blue light. (**A**) Schematic illustration of the vectors used in these experiments. (**B**) The pOPTO-BLUE-V1 and -V2 plasmids were transfected into HEK293-T cells for 8 h. Then, one experimental group was kept in the dark, while another group of cells were exposed to blue light for 16 h. After that, a Western blot was performed to detect EGFP protein levels. GADPH was used as a loading control. (**C**) EGFP protein levels were quantified by densitometry as indicated in the material and methods section. The intensity of EGFP bands was normalized with the intensity of GAPDH bands. Different asterisks (*) indicate significant differences between dark and light conditions. A two-way ANOVA test following Bonferroni’s post-hoc test was performed (* *p* < 0.05 and ** *p* < 0.01).

**Figure 3 ijms-24-09537-f003:**
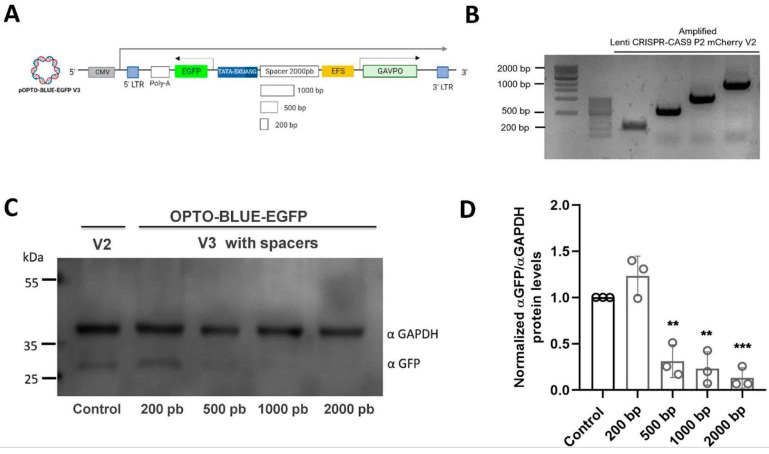
Evaluation of EGFP expression in darkness using different spacers between the promoters of the pOPTO-BLUE vector (UAS and EFS) in HEK293-T cells. (**A**) Shows the schematic illustration of the pOPTO-BLUE-GFP-V3 lentiviral vector with different versions of spacers. (**B**) PCR amplification from the plentiCRISPR-CAS9-P2-mCherry-V2 was performed to obtain amplicons of different sizes (2000 bp, 1000 bp, 500 bp, and 200 bp). In (**C**), HEK293-T cells were transfected with the pOPTO-BLUE-EGFP vector + spacers and kept in darkness for 24 h. Subsequently, cells were homogenized to detect and quantify by Western blot the expression of EGFP and GAPDH (as load control) using specific antibodies. (**D**) EGFP protein levels were quantified by densitometry, as indicated in the material and methods section. The intensity of EGFP bands was normalized with the intensity of GAPDH bands. Different asterisks (*) indicate significant differences between dark and light conditions. A two-way ANOVA test following Bonferroni’s post hoc test was performed (** *p* < 0.01 and *** *p* < 0.001).

**Figure 4 ijms-24-09537-f004:**
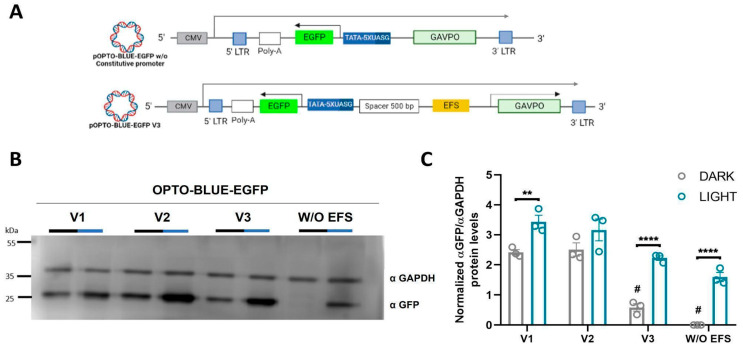
Blue light-mediated EGFP expression is induced by the different versions of the pOPTO-BLUE-EGFP (V1, V2, V3, and V2_W/O EFS) lentiviral vector in HEK293-T cells. (**A**) Schematic illustrations of the pOPTO-BLUE-EGFP lentiviral vector with a 500 bp spacer (pOPTO-BLUE-EGFP-V3_500bp) and another version of the pOPTO-BLUE-EGFP-V2 plasmid without constitutive EFS promoter (W/O) are shown. (**B**) Different versions of the pOPTO-BLUE-EGFP lentiviral vector (V1, V2, V3, and V2_W/O) were transfected into HEK293-T cells for 8 h. Then, one experimental group was kept in darkness, while another experimental group of cells was exposed to blue light for 16 h. Finally, a Western blot was performed to detect EGFP protein levels. GADPH was used as a loading control. (**C**) As indicated in the material and methods section, EGFP protein levels were quantified by densitometry. The intensity of EGFP bands was normalized with the intensity of GAPDH bands. Different asterisks (*) indicate significant differences between dark and light conditions, and pound sign (#) indicates significant differences between dark conditions with different plasmid versions. A two-way ANOVA test following Bonferroni’s post hoc test was performed (** *p* < 0.01 and **** *p* < 0.0001) (# *p* < 0.001).

**Figure 5 ijms-24-09537-f005:**
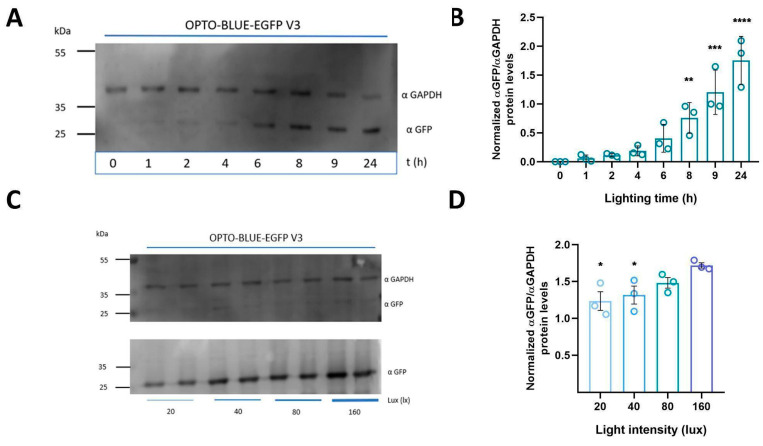
Time course and intensity determination of light-switchable EGFP expression using pOPTO-BLUE-EGFP-V3 vector in HEK293-T cells. (**A**) HEK293-T cells transfected with the pOPTO-BLUE-EGFP-V3_500bp plasmid were illuminated with blue light to induce EGFP expression at different times (0, 1, 2, 4, 6, 8, 9, and 24 h). The expression of EGFP and GAPDH was evaluated by immunodetection, and quantification is shown in (**B**). (**C**) Different blue illumination intensities (20, 40, 80, and 160 lux) were evaluated to determine the appropriate parameters to obtain the highest induction efficiency with pOPTO-BLUE-V3_500bp. For this, HEK293-T cells were transfected with the pOPTO-BLUE-EGFP-V3_500bp plasmid, then different groups of cells were illuminated with blue light at different intensities to induce EGFP expression. The expression of EGFP and GAPDH was evaluated by immunodetection (**D**) EGFP protein levels were quantified by densitometry, as indicated in the material and methods section. The intensity of EGFP bands was normalized with the intensity of GAPDH bands. Different asterisks (*) indicate significant differences between dark and light conditions as determined using a one-way ANOVA test following Dunnet’s post-hoc test (* *p* < 0.05, ** *p* < 0.01, *** *p* < 0.001, **** *p* < 0.0001).

**Figure 6 ijms-24-09537-f006:**
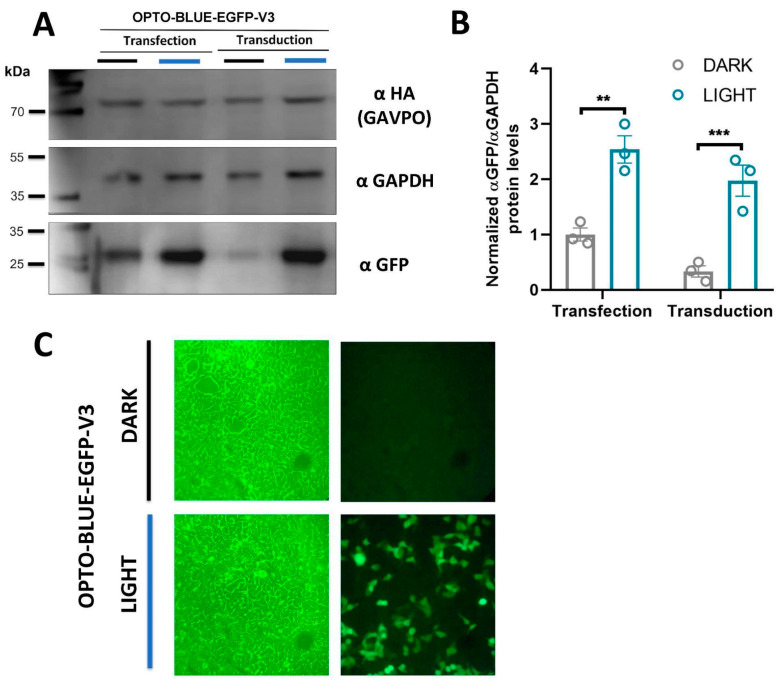
Blue-light inducible expression of EGFP in HEK293-T cells transfected/transduced with the lentivirus OPTO-BLUE-EGFP-V3_500bp. For this, HEK293-T cells were transduced with the OPTO-BLUE-GFP-V3_500bp lentivirus for 48 h. Then, one experimental group was kept in darkness, while another group of cells was exposed to blue light for 16 h. (**A**) Immunodetection was performed against EGFP and GAPDH (loading control), and quantification of the EGFP protein band intensity was performed, compared to the GAPDH protein band intensity, used as the loading control (**B**). Finally, 40× magnification epifluorescence microscopy reveals EGFP expression in cells transduced with the OPTO-BLUE-EGFP-V3_500bp lentivirus and exposed to blue light (**C**). Different asterisks in the graph indicate significant differences between dark and light conditions. A two-way ANOVA test following Bonferroni’s post-hoc test was performed (** *p* < 0.01, *** *p* < 0.001).

## Data Availability

The original data of this present study are available from the corresponding author.

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
