# Peer review of "OPTO-BLUE: An Integrated Bidirectional Optogenetic Lentiviral Platform for Controlled Light-Induced Gene Expression"

_ijms, 2023, doi:10.3390/ijms24119537_

Round 1
Reviewer 1 Report
In a recent paper published in Electronic Journal of Biotechnology (2021) the group of Zamorano adapted the LightOn system to a lentiviral system based on co-transduction. Here, the OPTO-BLUE system is presented carrying all components of the LightOn system on a single lentiviral plasmid to simplify the use of the system. Although OPTO-BLUE may has advantages, the data shown in the manuscript are incomplete and not convincing. Some points are mentioned:
1. Expression levels of GAVPO have to be shown.
2. Fig. 4 B/C: Explain the light-induced expression of EGFP without GAVPO expression, see text line 198.
3. Fig. 5 A/B: For the time course experiment, cells were exposed to blue light for several time points (1 to 24 hours) starting eight hours after transfection. This could mean that the expression of the reporter EGFP correlates with an increased background activity of accumulated GAVPO independent of light exposure.
4. Fig. 6 C: This figure is missing.
5. Supplementary Materials: Table S1 and Video S1 are missing.
6. Line 25: In the abstract it is mentioned that cells were “exposed to cycles of blue light illumination”. These data are not included in this manuscript.
7. Lines 338/339: It seems that a draft of the manuscript has been submitted.
Author Response
Reviewer 1
(x) I would not like to sign my review report
( ) I would like to sign my review report
Quality of English Language
(x) I am not qualified to assess the quality of English in this paper
( ) English very difficult to understand/incomprehensible
( ) Extensive editing of English language required
( ) Moderate editing of English language
( ) Minor editing of English language required
( ) English language fine. No issues detected
|
Yes |
Can be improved |
Must be improved |
Not applicable |
|
|
Does the introduction provide sufficient background and include all relevant references? |
( ) |
( ) |
(x) |
( ) |
|
Are all the cited references relevant to the research? |
( ) |
(x) |
( ) |
( ) |
|
Is the research design appropriate? |
( ) |
(x) |
( ) |
( ) |
|
Are the methods adequately described? |
( ) |
( ) |
(x) |
( ) |
|
Are the results clearly presented? |
( ) |
( ) |
(x) |
( ) |
|
Are the conclusions supported by the results? |
( ) |
( ) |
(x) |
( ) |
Comments and Suggestions for Authors
In a recent paper published in Electronic Journal of Biotechnology (2021) the group of Zamorano adapted the LightOn system to a lentiviral system based on co-transduction. Here, the OPTO-BLUE system is presented carrying all components of the LightOn system on a single lentiviral plasmid to simplify the use of the system. Although OPTO-BLUE may has advantages, the data shown in the manuscript are incomplete and not convincing. Some points are mentioned:
- Expression levels of GAVPO have to be shown.
The expression level of GAVPO was shown in Figure 6A using an anti-HA antibody. The expression levels of GAVPO in both transfection and transduction experiments are presented. To provide better clarity to the readers, we have changed the label in Figure 6A to HA (GAVPO).
- Fig. 4 B/C: Explain the light-induced expression of EGFP without GAVPO expression, see text line 198.
We greatly appreciate the reviewer's comment on this point. In transfection experiments using the pOPTO-BLUE-EGFP plasmid, GAVPO expression is still possible due to the CMV promoter. Under these conditions, no background expression of EGFP is observed, suggesting that the EFS promoter causes the background expression due to its proximity to the UAS minimal promoter. The paragraph in lines 206-208 has been modified in the revised version of the manuscript to address this issue.
- Fig. 5 A/B: For the time course experiment, cells were exposed to blue light for several time points (1 to 24 hours) starting eight hours after transfection. This could mean that the expression of the reporter EGFP correlates with an increased background activity of accumulated GAVPO independent of light exposure.
We appreciate the valuable comment regarding a potential GAVPO-independent expression of EGFP. Figure 3C demonstrates that negligible expression of EGFP is observed after keeping the cultures in the dark for 24 hours post-transfection. Similarly, in Fig. 5A, we maintained the cultures in the dark for 8 hours after exposure to blue light, with no observed expression of EGFP. However, in the presence of blue light, the accumulation of EGFP is observed over time, up to 36 hours post-transfection (24 hours of induction). We apologize for any confusion caused by the lack of clarity in our manuscript regarding the transfection times before induction. We have made the necessary changes to ensure that this information is explicitly stated in the revised manuscript.
- Fig. 6 C: This figure is missing.
We apologize for this oversight. Unfortunately, as the reviewer mentioned, a draft version of the manuscript was mistakenly uploaded, which did not include Figure 6C. We are submitting a new revised version of the manuscript with the correct figure and figure legend for Figure 6. In addition, a new section in material and methods has been added to provide methodological information about panel C in the manuscript.
- Supplementary Materials: Table S1 and Video S1 are missing.
We apologize for the omission of Table S1 and Video S1. Due to an inadvertent error, this section remained from the original template. We have corrected this mistake in the revised version of the manuscript.
- Line 25: In the abstract it is mentioned that cells were “exposed to cycles of blue light illumination”. These data are not included in this manuscript.
After conducting preliminary experiments with lighting cycles to avoid overheating the cell incubator, we determined that LED lights had no impact on the temperature. Therefore, we proceeded with continuous lighting exposure for our experiments. We apologize for the confusion.
- Lines 338/339: It seems that a draft of the manuscript has been submitted.
We acknowledge and appreciate the reviewer's observation. We sincerely apologize for the fact that the last draft we sent still contained some comments and corrections. We assure you that this new resubmission is free from errors or comments.

Reviewer 2 Report
Revision ijms – 2402390
The study by Arancibia et al. aimed to optimize the LightON system, an optogenetic tool that regulates the expression of a gene of interest using blue light as an inducer. The authors assembled all components of the LightON system into a single lentiviral plasmid, creating the OPTO-BLUE system.
They functionally validated the system using enhanced GFP fluorescent protein as an expression reporter and evaluated its efficiency in HEK293-T cells exposed to cycles of LED blue light illumination. The results demonstrated that the optimized OPTO-BLUE system has the potential to selectively modulate gene expression of a protein by blue light, providing enhanced spatio-temporal resolution.
The study was well designed, the experiments were well conducted, and the analyses/results are convincing. The manuscript is clear and well-written. I have only a few minor comments for the authors:
(1) Two parametric tests are introduced as part of the plan of data analysis. Please report ANOVA F-value and degree of freedom. There is not space limitation for methods and results, so why not including this information in the main manuscript? This information will help convince the reader about the significance of the results, particularly with a small sample size;
(2) In FIG. 5D, it appears that the control of expression (without light stimulation) is missing;
(3) In FIG. 6B, please represent all data points. There is any statistical significance comparing dark protocols? In line 273, it is introduced a panel (C) not included;
(4) The discussion section would benefit from a clear introductory paragraph that summarizes the main achievements of the study. In addition, the conclusion paragraph could be improved by including a statement about the limitations of the technique, future questions that need to be addressed, or potential translational applications;
(5) Regarding the supplementary data, please include a legend for Fig. S1. Be aware that is not appropriate run any statistical analysis with a single point per contingency.
Author Response
Reviewer 2
( ) I would not like to sign my review report
(x) I would like to sign my review report
Quality of English Language
( ) I am not qualified to assess the quality of English in this paper
( ) English very difficult to understand/incomprehensible
( ) Extensive editing of English language required
( ) Moderate editing of English language
( ) Minor editing of English language required
(x) English language fine. No issues detected
|
|
Yes |
Can be improved |
Must be improved |
Not applicable |
|
Does the introduction provide sufficient background and include all relevant references? |
(x) |
( ) |
( ) |
( ) |
|
Are all the cited references relevant to the research? |
(x) |
( ) |
( ) |
( ) |
|
Is the research design appropriate? |
(x) |
( ) |
( ) |
( ) |
|
Are the methods adequately described? |
( ) |
(x) |
( ) |
( ) |
|
Are the results clearly presented? |
( ) |
(x) |
( ) |
( ) |
|
Are the conclusions supported by the results? |
( ) |
(x) |
( ) |
( ) |
Comments and Suggestions for Authors
The study by Arancibia et al. aimed to optimize the LightON system, an optogenetic tool that regulates the expression of a gene of interest using blue light as an inducer. The authors assembled all components of the LightON system into a single lentiviral plasmid, creating the OPTO-BLUE system.
They functionally validated the system using enhanced GFP fluorescent protein as an expression reporter and evaluated its efficiency in HEK293-T cells exposed to cycles of LED blue light illumination. The results demonstrated that the optimized OPTO-BLUE system has the potential to selectively modulate gene expression of a protein by blue light, providing enhanced spatio-temporal resolution.
The study was well designed, the experiments were well conducted, and the analyses/results are convincing. The manuscript is clear and well-written. I have only a few minor comments for the authors:
- Two parametric tests are introduced as part of the plan of data analysis. Please report ANOVA F-value and degree of freedom. There is not space limitation for methods and results, so why not including this information in the main manuscript? This information will help convince the reader about the significance of the results, particularly with a small sample size.
We appreciate the suggestion. In the revised version of the main manuscript, we have added the F-value, degrees of freedom, and p-values in the results section. We have also included information to specify whether a one- or two-way ANOVA test was used in the figure legends.
- In FIG. 5D, it appears that the control of expression (without light stimulation) is missing.
The standard light intensity used in all our experiments was 160 lux. In this particular experiment, we aimed to evaluate whether the induction of EGFP expression could be tunable by decreasing the intensity of blue light, resulting in lower EGFP expression levels compared to the standard intensity used in the experiments (160 lux). Figure 4D demonstrates significant reductions in EGFP expression at low illumination intensities (20 and 40 lux). We apologize for the lack of clarity in explaining this experiment and have made the necessary changes to the manuscript to provide a better understanding for the reader.
3.- In FIG. 6B, please represent all data points. There is any statistical significance comparing dark protocols? In line 273, it is introduced a panel (C) not included.
We appreciate this observation. All the data points are now visible in graph 6B. However, despite the apparent trend observed in the blot, the statistical analysis for the dark conditions did not yield a significant decrease after performing an ANOVA test and comparing all the experimental groups. However, a significant decrease in EGFP expression was observed, analyzed by Student's t-test. We have made the changes in the manuscript to clarify this point. Additionally, we have added the missing panel C in Figure 6, as mentioned in the manuscript. We have also provided methodological information about panel C in the manuscript. We apologize for this oversight, as Figure 6 was part of a draft manuscript that was mistakenly uploaded.
Student’s t-test Dark Transfection vs Dark Transduction. Data from Figure 6. *p < 0.05.
(4) The discussion section would benefit from a clear introductory paragraph that summarizes the main achievements of the study. In addition, the conclusion paragraph could be improved by including a statement about the limitations of the technique, future questions that need to be addressed, or potential translational applications.
We appreciate your valuable suggestions. It aligns perfectly with our vision, and we have taken actions to enhance the discussion by adding a paragraph that summarizes the main achievements, as suggested by the reviewer. We are confident that these improvements will effectively showcase the significance of our research. Thank you again for your input.
(5) Regarding the supplementary data, please include a legend for Fig. S1. Be aware that is not appropriate run any statistical analysis with a single point per contingency.
The legend for Figure S1 has been included. The previous graph has been updated to the original double point per contingency, without statistical analysis.

Round 2
Reviewer 1 Report
The new resubmitted version of the manuscript is now complete and all questions and remarks of this reviewer were considered.
The OptoBlue system is based on the expression of GAVPO under the control of a constitutive promoter and its blue light-dependent dimerization and activation. Thus, for completeness the expression of GAVPO should be shown in the experiments as done in Fig. 6A for transfected and transduced cells. At least in Fig. 4 it would be interesting to see the expression level of GAVPO under control of the EFS promoter compared to the construct lacking the EPS promoter leading to CMV promoter driven GAVPO expression.
Minor point:
Legend to Fig. 2, line 151: “transfected” not “co-transfected”.
Author Response
Dear Reviewer 1,
We sincerely appreciate your insightful comments and the opportunity to address them. We have carefully reviewed your suggestions and would like to provide further clarification regarding the points you raised.
Regarding Figure 4, our primary objective was to investigate the source of background expression of EGFP in dark conditions within the OptoBlue system. We hypothesized that the proximity of the constitutive EFS promoter, positioned in a head-to-head CIS configuration, might be influencing the activation of the minimal UAS promoter. The results presented in Figure 4 indeed support this hypothesis, as we observed a correlation between increased distance between the promoters and decreased EGFP expression in the darkness. To provide additional evidence implicating the EFS promoter in the background expression of EGFP, we generated a construct without the EFS promoter. The complete absence of EGFP expression in darkness by this construct further supports the notion that the internal EFS promoter is influencing the background EGFP expression driven by the minimal UAS promoter.
We acknowledge your suggestion to explore the levels of GAVPO expression and its potential influence on the background EGFP expression. However, we did not delve deeper into this aspect due to several reasons specific to our experimental conditions. Firstly, in transfection experiments, a full packaging transcript can be expressed in the absence of the internal EFS promoter, as demonstrated in a similar way in previous studies (doi: 10.1128/JVI.72.10.8150-8157.1998). Therefore, we assumed a significant expression of GAVPO to be present in the transfection experiments for all four constructs used. Secondly, the minimal intracellular concentration of GAVPO required for activating the UAS minimal promoter after GAVPO dimerization remains unknown. Lastly, we attempted to shed light on this aspect in Figure 6, which provided meaningful data by measuring GAVPO expression in transfection experiments where GAVPO expression was triggered by both the CMV and EFS promoters. Interestingly, the levels of GAVPO expression under this condition were similar to the expression achieved solely by the EFS promoter in transduction experiments. Notably, in both conditions, the expression levels of EGFP after light induction were comparable, suggesting that the levels of GAVPO in both experimental scenarios are sufficient for maximal EGFP expression.
We believe that the points outlined above contribute to explaining the observed background activation in our constructs, which is an important aspect influencing the utility of the OptoBlue system as a light-inducible expression lentiviral platform. While we acknowledge the importance of determining the minimal intracellular concentration of GAVPO required for UAS promoter activation, we recognize that this investigation exceeds the scope of our current study. Nonetheless, we are committed to addressing this question in future research endeavors.
We would also like to address our concern about repeating the immunoblots in Figure 4 and determining the GAVPO protein levels. We estimate that this would require an additional six weeks. If the Editors deem it necessary to repeat this experiment, we kindly request your understanding and support in granting us the required time to respond to this request.
Once again, we sincerely appreciate your valuable feedback and the opportunity to improve our manuscript. Your input has undoubtedly strengthened the quality and clarity of our research. We eagerly await your further evaluation and guidance.
The minor point regarding the Legend to Fig. 2, line 151 has been corrected.
Once again, we sincerely appreciate your valuable feedback and the opportunity to improve our manuscript. Your input has undoubtedly strengthened the quality and clarity of our research. We eagerly await your further evaluation and guidance.
Sincerely,
Pedro Zamorano Ph.D.
